# Capillary Western Immunoassay Optimization of Estrogen Related Factors in Human Subcutaneous Adipose Tissue

**DOI:** 10.3390/mps5020034

**Published:** 2022-04-15

**Authors:** Jessica L. Hill, Kara B. McIver, Kaleigh Katzer, Michelle T. Foster

**Affiliations:** Department of Food Science and Human Nutrition, Colorado State University, Fort Collins, CO 80523, USA; jessica.lynn.hill@colostate.edu (J.L.H.); kara.mciver@colostate.edu (K.B.M.); kkatzer@rams.colostate.edu (K.K.)

**Keywords:** lipedema, adipose tissue, estrogen, estrogen receptor, simple wes, western blot, protein detection

## Abstract

Lipedema is a multifaceted chronic fat disorder characterized by the bilateral and disproportionate accumulation of fat predominantly in the lower body regions of females. Research strongly supports that estrogen factors likely contribute to the pathophysiology of this disease. We aim to help demonstrate this link by quantifying estrogen factor differences between women with and without lipedema. For time and lipedema adipose tissue conservation, the Protein Simple WES machine will be utilized in place of traditional western blotting. Here, we are interested in evaluating estrogen related factors, such as, but not limited to, estrogen receptors and enzymes involved in the successive conversions of cholesterol and androgens to estrogens in human subcutaneous adipose. Evaluation of these factors within adipose tissue, however, is novel for this instrument. Thus, we optimized tissue lysis and protein extraction for 11 proteins of interest. Antibodies and their working concentrations were determined based upon specific and distinguishable (signal-to-noise) peaks from electropherogram outputs across different tissue lysate concentrations. We found that overnight acetone precipitation proved to be the best procedure for extracting protein from lipid rich adipose tissue samples. Six of the eleven proteins were found to migrate to their expected molecular weights, however, five did not. For proteins that did not migrate as expected, overexpression lysates and empty vector controls were used to validate detection antibodies. Protein extract from subcutaneous adipose tissue and overexpression lysates were then combined to understand if migration was specifically altered by adipose tissue. From these results, we concluded that the lipid rich nature of adipose tissue in combination with the separation matrix designated for use with the WES were preventing the appropriate migration of some proteins rather than non-specific antibody binding or inappropriate preparation methods.

## 1. Introduction

Lipedema is a chronic fat disorder characterized by the bilateral and disproportionate accumulation of fat predominantly in the lower body region [1], yet it also occurs in upper body areas such as the arms and abdomen [2]. This symmetrical adipose tissue accumulation can gradually develop palpable fat nodules (lipomas) [1,2,3], alter the shape of the limb and texture of the skin [1,2,3], and become susceptible to bruising due to reduced vascular integrity [1,2,3]. Occurring primarily in females, estimates predict that lipedema affects 11–19% of this population [4]. However, these numbers may underestimate the true prevalence of disease given its frequency of misdiagnosis and late diagnosis [1,2,3]. Lipedema primarily occurs during puberty, but for others may appear or become exacerbated during other times of hormonal shift, such as pregnancy, menopause, or hormonal contraceptive use [1,5]. Lipedema is scarcely documented in males, but when it occurs it is associated with low testosterone and/or increased estrogen levels [4]. Regardless, patient family histories support that lipedema is a heritable disease [2,6] and research shows an X-linked dominant pattern of inheritance, or more likely, an autosomal dominant inheritance with sex limitation [7].

Lipedema disorder is associated with many pathophysiological outcomes. This classically includes excessive adipose tissue accumulation. Despite differences in fat deposition and timing of fat accumulation, lipedema is commonly mistaken for general obesity. Shared features among adipose tissue from lipedema patients—even those who are not obese—and adipose tissue from obese patients include hypertrophic adipocytes, macrophage infiltration, and crown like structures surrounding adipocytes [8]. However, important distinctions can also be drawn. Ishaq et al. differentiate adipose tissue from lipedema patients and their BMI-matched controls through alterations in signaling, pathways regulating cell function, and adipocyte-derived stem cell proliferation [9]. In addition to fat accumulation, swelling can occur in limbs. Hence, lipedema is also often mistaken for lymphedema, an impairment of the lymphatic system [4,6,10]. Pain and tenderness are additionally reported to occur in limbs afflicted by lipedema [3]. Some postulate this is caused by factors like inflammation, tissue damage, fibrosis and nerve fiber abnormalities [3,10,11,12], edema, and compressed small blood vessels [3,10,11,12]. The pathophysiology of lipedema led to a call for preventive measures, effective treatment, even a cure, and yet the underlying etiology remains unclear.

Lipedema likely has a multifaceted etiology, yet research strongly supports that estrogen contributes to the pathophysiology of the disease [1,5]. Estrogen has been previously demonstrated to play a role in adipose tissue accumulation and distribution [3,11], fibrosis, and inflammation [3,11]. There are three primary forms of estrogen: estrone (E1), estradiol (E2), and estriol (E3). Although predominantly produced in the ovaries, corpus luteum, and the placenta [13], estrogen synthesis also occurs in adipose tissue [14,15,16]. Estrogen signaling through two nuclear hormone receptors, ERα and ERß, with ERα being the predominate form [11,17,18,19], is shown to influence adipose tissue lipogenesis, lipolysis and health in both human and mouse adipose depots [11,17,18,19]. Adipose tissue-derived estrogens contribute to overall circulating hormone concentrations [20,21] and act locally as an autocrine factor [22], in pre- and post-menopausal women [23,24].

We aim to demonstrate a link between estrogen and lipedema by quantifying estrogen factor differences between the adipose tissue of women with and without lipedema. We postulate this may contribute to the development of identification methods, prevention, and treatment options. Primary factors of interest include: (1) the two nuclear hormone receptors, ERα and ERß; (2) two adrenergic receptors, ARα2 and ARß2; and (3) the following conversion enzymes: p450c17, p450Aromatase, HSD17ß1, HSD17ß2, HSD17ß4, and HSD17ß7. ERα and ERß regulate adipose tissue metabolism, as well as lipid accumulation and loss [13,25,26]. The adrenergic receptors regulate adipose tissue accumulation and lipolysis and are of interest because both estrogen receptors are demonstrated to contribute to their function [16,27,28,29,30,31]. ERα activation is demonstrated to increase αAR expression [11,32], which promotes anti-lipolytic activity and subsequently lipid accumulation [11,17,18,19], whereas βARs stimulate lipolysis (Figure 1). The p450 enzymes and the HSD enzymes comprise parts of the steroid hormone synthesis pathways that influence the production of estrone (E1), estradiol (E2), and estriol (E3). All steroid synthesis pathways begin with cholesterol, the common precursor molecule, and are completed through successive enzymatic conversions from androgens to testosterone and eventually to estrogen [13,25,26] (Figure 1).

Western blot is an accepted, semi-quantitative method for identifying protein presence and abundance in a source. However, it is not without its flaws, including time investment [33], lack of reproducibility, quantity of sample and reliability issues [34]. For these reasons, the WES by Protein Simple (ProteinSimple, San Jose, CA, USA) was utilized for adipose tissue antibody verification. The WES system utilizes proprietary CE-SDS (capillary electrophoresis-sodium dodecyl sulfate) fluorescent tagging. CE-SDS replaces the traditional SDS-PAGE (sodium dodecyl sulphate–polyacrylamide gel electrophoresis) and has previously been found to produce comparable molecular weights for proteins with little or no post-translational modification [35]. Within individual capillaries, the WES automates protein loading, separation by size, blocking, washing, and detection, as opposed to traditional Western blotting, which requires each of these steps to be completed manually. Capillaries draw up samples, primary and secondary antibodies, blocking solutions, and wash buffers in the appropriate order from a hand-prepared plate. Through this automation, the WES can improve reproducibility and reliability compared with traditional methods in hours rather than the days traditionally necessary for results.

Automated capillary electrophoresis technology has been validated in various cell lysate [34,36] and tissue sample types, including human adipose tissue [37] and lipid-rich brain tissue [38]. However, to our knowledge, quantification of estrogen related factors from adipose tissue have yet to be tested on this platform. As such, it was our aim to develop appropriate preparation methods and validate antibodies for the quantification and identification of 11 estrogen signaling and synthesis factors of interest in human, female subcutaneous adipose tissue before moving on to analyzing samples from lipedema patients [39]. We hypothesized that the WES system would be an appropriate platform to study 11 estrogen-related synthesis and signaling proteins in human female SAT based upon existing research utilizing the WES as a reasonable substitute for traditional Western blotting [33,34,35,36,37,38,39].

## 2. Materials and Methods

### 2.1. Protein Extraction and Quantification

Protein was extracted from snap frozen human female subcutaneous adipose tissue (CSI20380A, Cell Sciences Newburyport, MA, USA) by treating with lysis buffer (150 mM NaCl, 1% Triton X-100, 10 mM Tris HCL pH 7.4, 5 mM EDTA, diH_2_O, protease (P8340, Sigma-Aldrich, St. Louis, MO, USA) and phosphatase (P0044, Sigma-Aldrich) inhibitor cocktails), homogenizing and then sonicating on ice for 15 s. Samples were then centrifuged at 16,000× *g* and 4 °C for 20 min, and the protein containing supernatant was removed. Excess lipid was extracted from sample supernatants using an overnight acetone precipitation at −20 °C. Sample protein concentrations were determined using total protein reagent (T1949; Sigma-Aldrich) for the Biuret method.

A total protein assay was conducted for data normalization and comparison on the WES per the manufacturer’s instructions with a 12–230 kDa separation module (SM-W004, ProteinSimple, San Jose, CA, USA) and total protein detection module (DM-TP01, ProteinSimple). Target proteins were detected with the following primary antibodies: ERα (8644, Cell Signaling Technology, Danvers, MA, USA); ERß (MA1-23217, Thermo Fisher Scientific, Waltham, MA, USA); HSD17ß1 (ab51045, Abcam, Boston, MA, USA); P450c17 (ab125022, Abcam); P450 Aromatase (ab124776, Abcam); SULTE1 (MAB5545-SP, R&D Systems, Minneapolis, MN, USA); HSD17ß2 (NBP2-01952, Novus Biologicals, Littleton, CO, USA); HSD17ß4 (VMA00380, Bio-Rad, Herceules, CA, USA); ARα2 (PA1-048, Invitrogen, Waltham, MA USA); ARß2 (ab182136, Abcam). The following anti-rabbit (DM-001 ProteinSimple detection module, San Jose, CA, USA) and anti-mouse (DM-002 ProteinSimple detection module, San Jose, CA, USA) secondaries were used. All antibodies were tested at a 1:10 and 1:50 concentration, paired with a 0.5 and 1.0 mg/mL protein concentration. No-sample and no-primary antibody controls were included to evaluate non-specific antibody binding with Simple WES reagents and/or non-specific antibody binding of the secondary antibody to the lysate. See Table 1 for an example plate layout. HSD17ß2 and HSD17ß7 were further tested at additional lysate dilutions (0.0625, 0.125, 0.25, 0.3, 0.4, and 0.5 mg/mL) to improve protein detection signal and migration.

The Compass software (version 5.0.1, Protein Simple, San Jose, CA, USA) provided with the WES instrument was used to process and analyze results. Compass produces a size-based analysis, displayed as bands in a lane view, like in traditional Western results—seen below in Figure 2 and Figure 3. Electropherograms, reported in all figures below, show quantitative distributions of results as peaks and allow for multiple samples to be overlain for comparison.

One additional sample preparation method (extraction kit, Minute™ total protein for adipose tissue (AT-022, Invent Biotechnologies, Plymouth, MN, USA) was tested as an effort to enhance lipid removal from samples compared with the overnight acetone precipitation method. This was done specifically for HSD17ß2 (expected: 42 kDa, observed: 61–63 kDa), HSD17ß7 (expected: 32–35, observed: 61–62 kDa), and SULTe1 (expected: 35, observed: 62–63 kDa), as we postulated that the aberrant migration of these antibodies may be related to incomplete lipid removal. Lysate concentrations remained at 0.5 and 1.0 mg/mL with 1:10 or 1:50 antibody concentrations.

### 2.2. Control Lysates

Positive control lysates, previously tested and verified [40,41,42,43,44] by traditional Western blot, were used to further verify antibody specificity on the WES system for five target proteins (HSD17ß1, HSD17ß2, HSD17ß4, HSD17ß7, and SULTe1) displaying aberrant migration patterns. Control lysates tested include those for: HSD17ß1 (human placental whole cell lysate; NB820-59248, Novus Biologicals), HSD17ß2 (human embryonic kidney cells; NBL1-11726, Novus Biologicals), HSD17ß4 (human embryonic kidney cells; NBL1-11728, Novus Biologicals), HSD17ß7 (human embryonic kidney cells; NBP2-07042, Novus Biologicals) and SULTe1 (human embryonic kidney cells; ab94289, Abcam). The stock concentration for the SULTe1 overexpression lysate was not provided by the manufacturer, thus serial dilutions were used (1:50, 1:100, 1:200, 1:400 and 1:800). The remaining four overexpression lysates were run as 0.5 mg/mL or 0.8 mg/mL. Antibody concentrations of 1:10 and 1:50 were tested, matching previous runs. Empty vector negative control lysates provided by the manufacturer were used.

To directly investigate the impact of lipid from adipose tissue protein extracts on target migration patterns, human female subcutaneous adipose tissue lysate was combined with all four of the overexpression control lysates tested. For each, a mixture of high (75:25) and low (25:75) control lysate to human adipose tissue was tested. All mixtures produced were tested with antibody at a 1:10 concentration, which produced the clearest peak and lowest signal-to-noise ratio in initial testing.

## 3. Results

Through multiple rounds of testing and validation, we found that the Simple WES capillary electrophoresis system was able to accurately capture 10 of the 11 proteins we hoped to identify in this trial. We were unable to accurately detect HSD17ß7 due to the lack of specificity of the antibody. Six of our target proteins migrated to their expected or previously reported molecular weights in initial testing and, therefore, lysate preparation methods were accepted. However, the other five target proteins did not migrate to their expected molecular weights and, thus, we undertook additional testing to troubleshoot. We theorized that the discrepancy between the expected and observed molecular weights was not a result of misidentification by detection antibodies but rather due to the nature of this sample type on the WES system.

### 3.1. Experiment 1–Optimization

Where possible, the molecular weights of our adipose tissue derived target proteins (i.e., ERα, HSD17ß1, p450 Aromatase) were compared with data from previous validation in other tissues on the WES machine [45,46,47]. However, when WES data was not readily available (i.e., ERß, ARα2, ARß2, p450c17, HSD17ß2, HSD17ß4, and SULTe1), traditionally derived molecular weights were used, which is not directly comparable [48,49,50,51,52,53]. ERα, ERß, and p450 Aromatase, have an observed weight of ~60 kDa via traditional Western blot. With WES these antibodies migrated within a reasonable distance of the expected molecular weights resulting in 66, 59, and 55 kDa [45,46,48] (Figure 2A,B,D). Therefore, we concluded that detection antibodies for ERα, ERß, and p450 Aromatase were specifically binding respective target proteins and thus acceptable for use [45,46,48]. P450c17 has been validated for a single peak at 57 kDa via traditional Western blotting, however in our subcutaneous adipose tissue trials, P450c17 showed three peaks at ~59, ~84 and ~151 kDa [49] (Figure 2C). We speculate that these three peaks have always existed, yet due to the increased sensitivity of the WES machine compared with traditional Western blot, detection of these isoforms is clearer. The peak at ~84 kDa was the smallest of the three, while peaks at both the lowest and highest observed molecular weights were three to four times larger. While ARß2 and ARα2 did not migrate to their predicted molecular weights of 49 and 45 kDa [45,46] (Figure 2E,F), others have previously reported similar findings via traditional Western blot [52,53]. As such, we concluded that these antibodies were specifically binding target proteins and acceptable for downstream use.

**Figure 2 mps-05-00034-f002:**
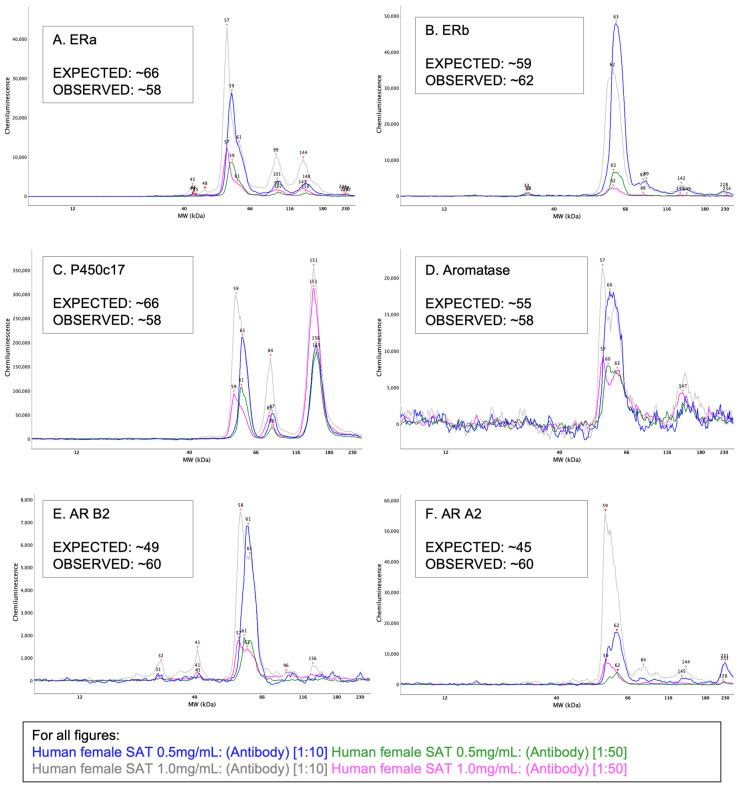
All electropherograms report lysate concentrations of 0.5 and 1.0 mg/mL with 1:10 and 1:50 antibody concentrations. These detection antibodies were identified within reasonable distance of their target protein’s predicted molecular weight, having a clean peak, and thus considered to be specifically binding target proteins of interest: (**A**) estrogen receptor α (**B**) estrogen receptor ß (**C**) p450c17 (**D**) p450Aromatase (**E**) adrenergic receptor ß2 and (**F)** adrenergic receptor α2.

All lysate–antibody concentration combinations of HSD17ß1 (expected: 35 kDa), HSD17ß2 (expected: 42 kDa), HSD17ß4 (expected: 79 kDa), HSD17ß7 (expected: 32 kDa), and SULTe1 (expected: 35 kDa) (Figure 3A–E) failed to migrate to previously described molecular weights [47,52,53,54,55].

**Figure 3 mps-05-00034-f003:**
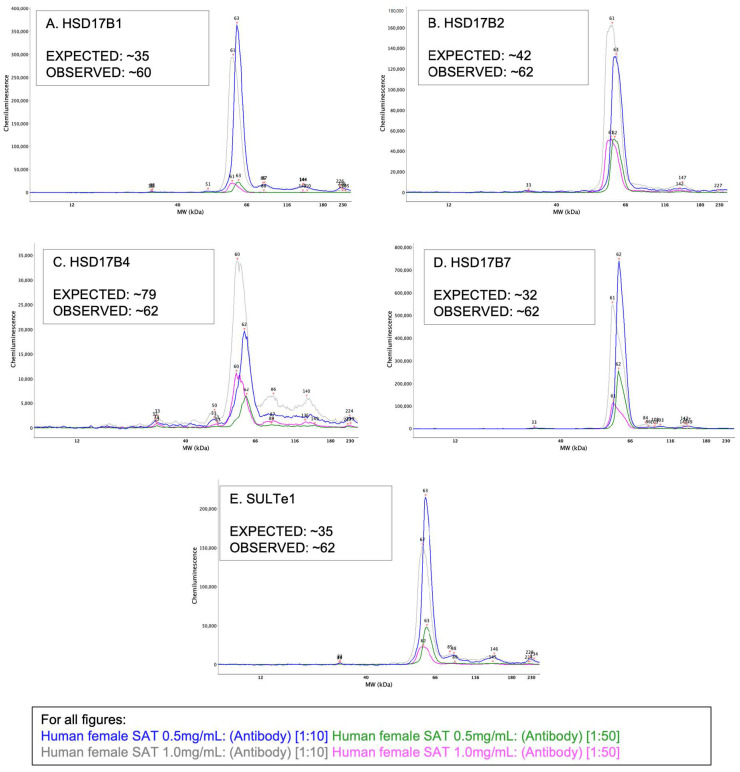
All electropherograms report lysate concentrations of 0.5 and 1.0 mg/mL with 1:10 and 1:50 antibody concentrations. These detection antibodies were not close to their target protein’s predicted molecular weight and were, thus, not considered to be specifically binding target proteins of interest: (**A**) HSD17ß1 (**B**) HSD17ß2 (**C**) HSD17ß4 (**D**) HSD17ß7 and (**E**) SULTe1.

### 3.2. Experiment 1.1–Lysate Reduction and Minute™ Kit Testing

We posited that reducing sample lysate concentrations or more thoroughly removing lipid from samples may contribute to improved SDS coating and protein migration through capillaries. Thus, a range of lowered protein concentrations and the Minute™ kit were applied, separately, to improve results.

A range of reduced sample lysate concentrations (0.0625, 0.125, 0.25, 0.3, 0.4, and 0.5 mg/mL) were tested with 1:10 and 1:50 antibody concentrations using both HSD17ß2 (Figure 4A) and HSD17ß7 (Figure 4B). In both cases, molecular weights remained nearer to those observed in Experiment 1 than to the expected weights, both falling at approximately 62 kDa rather than 42 and 32 kDa, respectively. Because this resulted in no change in the observed molecular weights of targets from adipose tissue, we did not test differing lysate concentrations on the remaining target proteins.

There is no standard protocol for extracting protein from adipose tissue for Western blot analysis [56]. Therefore, we tested an alternative method for lipid extraction to the original overnight acetone precipitation on three of the proteins of interest with unexpected migration: HSD17ß2, HSD17ß7, and SULTe1. Figure 5 compares the molecular weights observed when utilizing each preparation, specifically 62 kDa for both methods and all three proteins. The kit did not alter the observed molecular weights in any meaningful way, thus, further testing with remaining antibodies was not conducted.

### 3.3. Experiment 2–Positive and Negative Control Lysates

To verify that detection antibodies were accurately capturing our target proteins, we tested them on positive and negative control lysates. Control lysates were obtained for the five target proteins observed at aberrant molecular weights in adipose tissue (HSD17ß1, HSD17ß2, HSD17ß4, HSD17ß7, and SULTe1). All HSD17ß antibodies were tested at a 0.5 mg/mL and 0.8 mg/mL lysate concentration paired with either 1:10 or 1:50 antibody concentrations. Because the stock concentration for the SULTe1 overexpression lysate was not provided by the manufacturer, serial dilutions were used (1:50, 1:100, 1:200, 1:400 and 1:800). Again, each paired with 1:10 and 1:50 antibody concentrations.

The detection antibody for HSD17ß1 has been validated in human placental tissue via traditional Western blot with a molecular weight of ~35 kDa [55]. As such, we verified HSD17ß1 detection and migration on the WES system in human placental tissue and found that it also migrated to ~35 kDa with one clean peak (Figure 6A). The antibody for HSD17ß2 was validated on the Wes using human embryonic kidney (HEK) over-expressor lysate [41]. In this lysate, HSD17ß2 migrated to its expected molecular weight of ~42 kDa (Figure 6B). A second peak at ~211–213 kDa was also detected, however, only in the 0.8 mg/mL lysate concentration. Detection and migration of HSD17ß4 was also verified in a HEK over-expressor lysate [42], and found to migrate to its predicted molecular weight of 76–78 kDa on the WES (Figure 6C). HSD17ß7 has an expected molecular weight of 32–35 kDa [43] and was found to migrate to only ~62 kDa in its HEK over-expressor lysate (Figure 6D). Last, SULTe1 migrated to ~39 kDa (Figure 6E), nearing its predicted molecular weight of ~35 kDa, in all dilutions of its over-expressor lysate [44]. For negative controls, empty vector lysates provided by the manufacturers were used.

### 3.4. Experiment 3–Control Lysates Combined with SAT Sample Lysates

Finally, we wanted to verify that sample lysates prepared from adipose tissue were causing aberrant protein migrations on the WES system. To do this, we combined a low (25:75) and high (75:25) quantity of lysate from female SAT with a low and high quantity of the over-expresser control lysates utilized in Experiment 2. We hypothesized that the pooled lysates would result in a “gradient” of molecular weights from what was observed with the over-expresser lysate, to the low-fat combination, to the high-fat combination, and finally the all-SAT lysate observed molecular weights.

All four pooled lysates revealed bimodal electropherograms, with one peak at or very near the “expected” molecular weight and another at or near the molecular weights observed in our previous experiments (Figure 7). Specifically, HSD17ß1 had peaks at ~31 kDa and ~60 kDa (Figure 7A). HSD17ß2 had peaks at ~44 kDa and ~61 kDa (Figure 7B). HSD17ß4 had peaks at ~61 kDa and ~78 kDa (Figure 7C). Finally, SULTe1 had peaks at ~37 kDa and ~60 kDa (Figure 7D). While results did not exactly reflect our hypothesis, they made clear that the antibodies were in fact capturing our desired target proteins and that the nature of our sample lysate obtained from adipose tissue was causing a shift in molecular weights. Table 2 summarizes the findings for all the described experiments.

Based on the combined results of the above experiments, we believe that the preparation methods and antibodies described here are effective and accurately capturing 10 of the 11 estrogen synthesis and signaling target proteins for this study using the Protein Simple WES system. We believe that the lipid rich nature of adipose tissue in combination with the reagents required for use with the WES system led to alterations in expected molecular weights but not the accuracy of this instrument.

## 4. Discussion

While there is speculation that the onset and progression of lipedema are linked to estrogen [1,3,57], studies showing quantifiable differences in estrogen-related protein abundance in SAT between patients and control groups have not yet been conducted. We aim to make inroads on this research by utilizing cutting-edge, rapid Western technology rather than traditional Western blot methods.

Research utilizing the Simple Western platform for automated Western blots is increasing [34]. However, literature describing the employment of this technology in relation to the presence of proteins in adipose tissue is scant [37]. We were unable to find any previous publications regarding validated antibodies for estrogen synthesis and signaling proteins in human subcutaneous adipose tissue. As such, we set out to determine effective preparation methods, lysate concentrations, and validate antibodies to identify and quantify our target proteins utilizing the WES machine.

Six of our proteins of interest (ERα, ERß, p450c17, p450Aromatase, ARß2, and ARα2) migrated within reasonable distance of their reported molecular weights based on traditional Western blotting. The remaining five proteins (HSD17ß1, HSD17ß2, HSD17ß4, HSD17ß7, and SULTe1), did not. We hypothesized that the lipid-rich nature of adipose tissue was affecting protein migration on the WES. To test this, we ran varying lysate concentrations and trialed different lysate preparation methods to improve protein migration and lipid removal from samples. We also verified antibody specificity in accurately detecting our target proteins using positive and negative control lysates.

In an initial test to determine if varying sample lysate concentrations would improve protein migration on the WES, we found that neither HSD17ß2 nor HSD17ß7 showed improved migration. We then tested an alternative sample lysate preparation method, the Minute™ Total Protein Extraction Kit for adipose tissue. We compared protein extraction and total yields, as well as sample migration on the WES, for sample lysates prepared using both the Minute™ Kit and overnight acetone precipitation methods. The Minute™ Kit combines a proprietary detergent free extraction buffer with a porous filter containing unique surface property and pre-defined pore size and thickness to separate oil from adipose tissue from the aqueous phase, allowing an un-biased representation of cellular proteins. Despite improvements in protein extraction and total yield with the Minute™ Kit, we did not see improvements in sample migration on the WES compared with our overnight acetone precipitation method. Acetone precipitation and purification of proteins from adipose tissue is an effective and reliable technique [58]. In an in-depth comparison of adipose tissue protein precipitation methods, Benabdelkamel et al. found that overnight precipitation in acetone was the most effective [58] and our results agree.

Five of our detection antibodies that captured proteins with aberrant migration patterns in subcutaneous adipose tissue (HSD17ß1, HSD17ß2, HSD17ß4, HSD17ß7, and SULTe1) on the WES were tested for specificity using positive and negative control lysates. We successfully validated the specificity of antibodies targeting HSD17ß1, HSD17ß2, HSD17ß4, and SULTe1, as determined by a single clean peak in positive but not negative control lysates. However, we found that the antibody selected for targeting HSD17ß7 was not specific and thus it was removed from our list and all downstream analysis.

Protein Simple, the WES manufacturer, has previously documented differences in apparent molecular weight in some proteins (“gel shifting”) based on several factors including matrix or gel type used in the procedure, running buffers, and some characteristics of the target proteins which can hinder complete, uniform sodium dodecyl sulfate (SDS) coating [59]. Characteristics impacting SDS coating include hydrophobicity and post-translational modifications [60,61]. Post-translational modifications known to cause gel-shifting include but are not limited to glycosylation, ubiquitination, and phosphorylation. Adipose tissue is particularly lipid rich and therefore could create hydrophobic interactions that inhibit proper migration on the WES system [59]. We believe that these influences are likely responsible for the altered protein migrations observed in our adipose tissue samples. Lu, Allred, and Jensen found a strong correlation between the quantification of CD38 and Erk proteins from adipose tissue cells with traditional Western blot and the WES system [35]. As a result, they deemed this capillary electrophoresis technology to be a “satisfactory alternative for analyses of human adipose tissue proteins” [35]. Others have similarly found the size-based capillary separation method comparable to traditional Western methods in human muscle, plasma, and cell extract [36,37,38,39,62]. This highlights the important differences between sample type run on the WES and overall run success.

It is also important to consider the proprietary nature of Simple Wes reagents as a limitation of the WES system and this study. A researcher’s ability to optimize protocols and methodologies stems from their complete understanding of a system. Therefore, without information regarding reagent composition we were somewhat limited in our ability to improve protein migration on the WES system. Future work should include the development of homebrew running buffers, allowing for better control of things like SDS concentration, in addition to modifying separation conditions on the WES. These and other alterations may lead to better sample migration and resolution.

Ultimately, we were able to optimize sample lysate and antibody concentrations, as well as tissue preparation methods, and validate 10 of our 11 antibodies targeting estrogen synthesis and signaling proteins. A limitation of this study, however, is that we only confirmed detection antibody specificity—using positive and negative control lysates—in “problematic” proteins (5 out of 11). For 9 of our target proteins, the 0.5 mg/mL lysate concentration in combination with the 1:10 antibody concentration was chosen for human Lipedema patient analysis based upon a strong and distinguishable signal. For p450c17, the 0.5 mg/mL lysate concentration in combination with the 1:50 antibody concentration was chosen. As we could not validate the specificity of our HSD17ß7 detection antibody, it was removed from current analysis. The overnight acetone precipitation method was chosen for sample lysate preparation given its effectiveness, ease, and low cost. Future studies should utilize this method to quantify differences in estrogen synthesis and signaling proteins between patients living with lipedema and age/BMI-matched controls, as well as between differently staged lipedema patients. Additional work should also characterize the potential implications of general obesity on protein analysis within adipose tissue using the WES system, as lipolysis and lipogenesis are altered in adipose tissue of lipedema patients. Nevertheless, our efforts here should translate seamlessly to samples from patients with general obesity.

## Figures and Tables

**Figure 1 mps-05-00034-f001:**
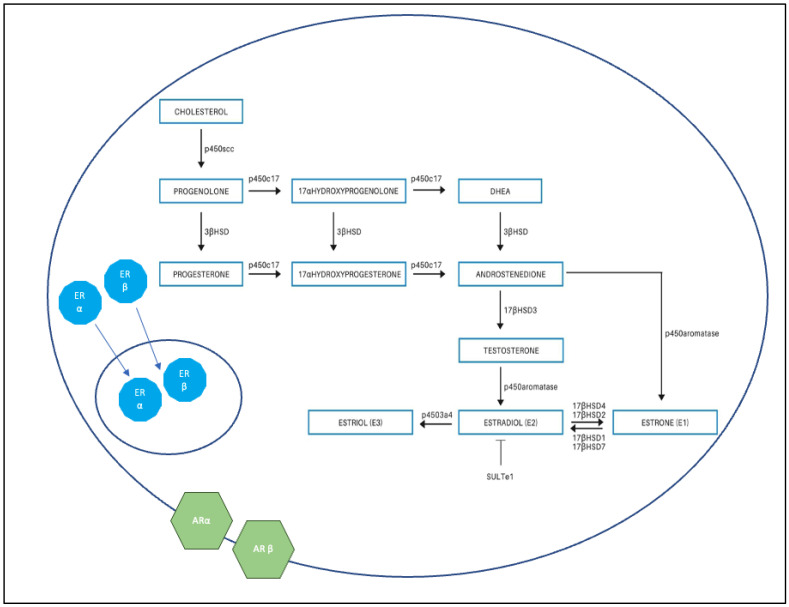
Conversion of androgens to estrogens in the cell. Estrogen receptor alpha (ERα) and beta (ERß) are nuclear hormone receptors located in the cytoplasm and upon ligand binding move into the nucleus. They regulate local adipose tissue metabolism. Adrenergic receptor alpha (ARa) and beta (ARß) are catecholamine receptors for the sympathetic nervous system and are localized to the cell membrane. AR2a negatively regulates lipolysis, whereas AR2ß positively regulates lipolysis. P450C17 catalyzes the conversion of pregnenolone and progesterone and is required to synthesize sex hormones. P450 Aromatase catalyzes the conversion of androgens to estrogens and is the rate-limiting enzyme of estrogen synthesis. 17ßHSD2 and 17ßHSD4 catalyze the conversion of E2 to E1. 17ßHSD1 and 17ßHSD7 catalyze the conversion of E1 to E2. Estrogen sulfotransferase (SULT1E1) sulfonates estrogens to inactive estrogen sulfates.

**Figure 4 mps-05-00034-f004:**
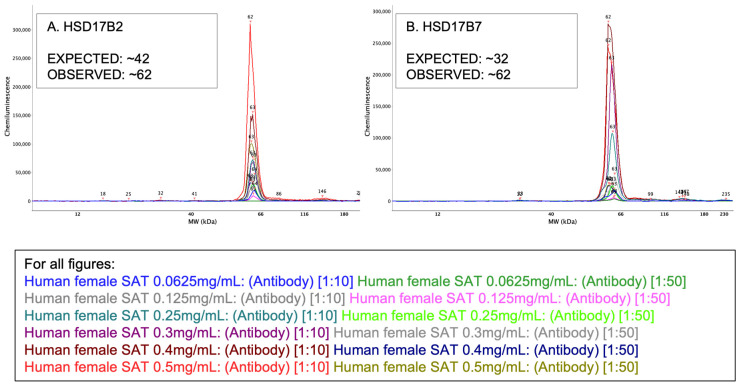
Protein quantification in reduced lysate concentrations (0.0625, 0.125, 0.25, 0.3, 0.4, and 0.5 mg/mL) paired with identical antibody concentrations (1:10 and 1:50) for (**A**) HSD17ß2 and (**B**) HSD17ß7.

**Figure 5 mps-05-00034-f005:**
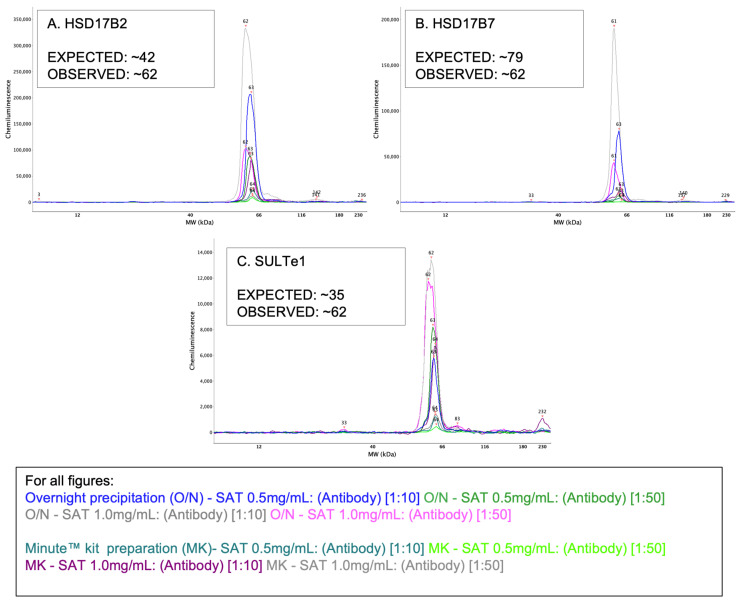
Each electropherogram compares results from overnight acetone precipitation and Minute™ kit preparation of SAT samples. Lysate concentrations of 0.5 and 1.0 mg/mL paired with 1:10 and 1:50 antibody concentration were tested: (**A**) HSD17ß2 (**B**) HSD17ß7 and (**C**) SULTe1.

**Figure 6 mps-05-00034-f006:**
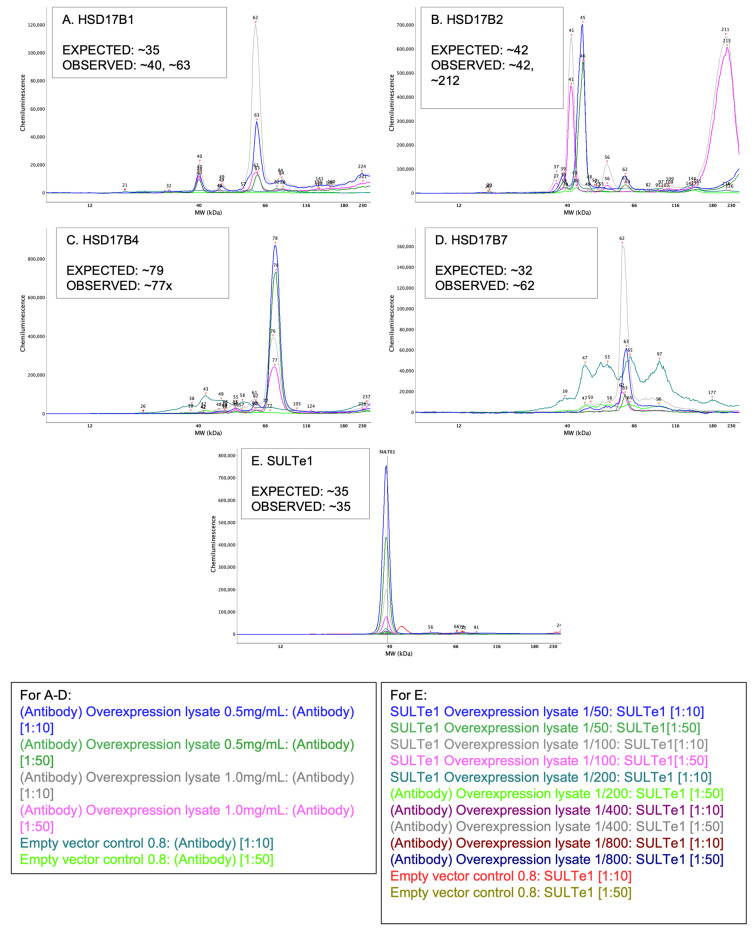
All electropherograms report results from over-expresser lysates at concentrations of 0.5 and 0.8 mg/mL (**A**–**D**) except for SULTe1 which was tested at: 1/50, 1/100, 1/200, 1/400, 1/800. Empty vector negative controls are also displayed on each electropherogram. Lysates were paired with antibody concentrations of 1:10 and 1:50. Data are listed as follows: (**A**) HSD17ß1, (**B**) HSD17ß2, (**C**) HSD17ß4, (**D**) HSD17ß7, and (**E**) SULTe1.

**Figure 7 mps-05-00034-f007:**
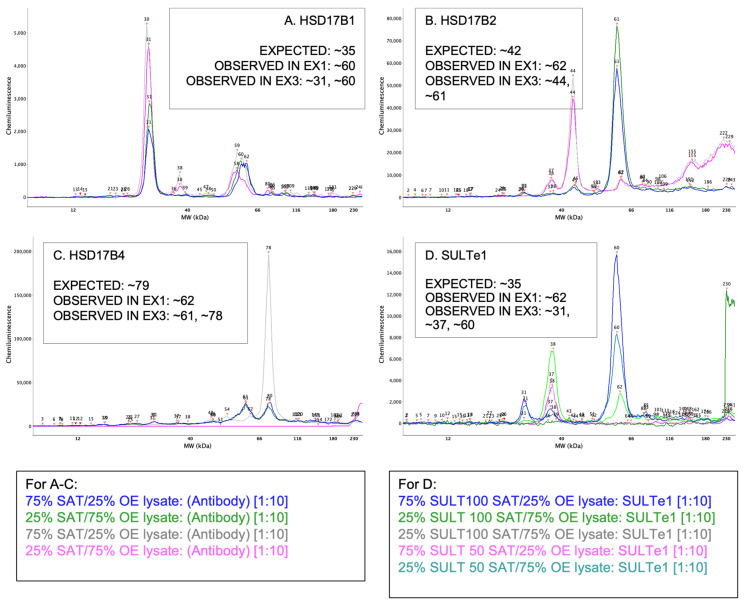
All electropherograms report data for high fat (75 SAT lysate:25 over-expresser lysate) and low fat (25 SAT lysate:75 over-expresser lysate) mixtures with 1:10 antibody concentrations in duplicate: (**A**) HSD17ß1 (**B**) HSD17ß2 (**C**) HSD17ß4 and (**D**) SULTe1.

**Table 1 mps-05-00034-t001:** Sample plate layout for the WES allowing us to determine optimal sample lysate and antibody concentrations, as well as verify antibody specificity. The plate layout designates six wells per sample with: a no-sample control, 0.5 mg/mL lysate—1:10 primary antibody, 1.0 mg/mL lysate—1:10 primary antibody, 0.5 mg/mL lysate—1:50 primary antibody, 1.0 mg/mL lysate—1:50 primary antibody, and a no-primary control.

Partial Simple Wes Plate Layout.								
	**1**	**2**	**3**	**4**	**5**	**6**	**7**	Repeat cells 2–7 in cells 8–25 with subsequent antibodies.
A	Biotinylated Ladder	0.1X Sample Buffer	0.5 mg/mL lysate	0.5 mg/mL lysate	1.0 mg/mL lysate	1.0 mg/mL lysate	1.0 mg/mL lysate	
B	Antibody Diluent 2 (AB D2)							
C	AB D2	ERα [1:10]	ERα [1:10]	ERα [1:50]	ERα [1:10]	ERα [1:50]	AB D2	
D	Streptavidin-HRP	Anti-Rabbit RTU (ready-to-use)						
E	Luminol-Peroxide							
F	Empty							
Notes		No sample control					No primary control	

**Table 2 mps-05-00034-t002:** Summary of expected and observed molecular weights for all 11 antibodies tested in each experiment.

Compiled Results of All Optimization Tests							
Antibody	Expected MW (kDa)	Observed MW ‘Initial Testing’	Observed MW ‘Reduced Lysate Concentrations’	Observed MW ‘Minute™ Extraction Kit’	Observed MW ‘Positive Control Lysate’	Observed MW ‘High-Fat Addition’	Observed MW ‘Low-Fat Addition’
ERα	66	58					
ERß	59	62					
ARα2	49	60					
ARß2	45	60					
p450c17	57	59, 84 & 151					
Aromatase	55	58					
HSD17ß1	35	60			49, 62	31, 60	31, 60
HSD17ß2	42	62		61	41	44, 61	44, 61
HSD17ß4	79	62			78	61, 78	61, 78
HSD17ß7	32	62	62	61	62		
SULTe1	35	62		61	35	31, 37, 60	31, 37, 60

## Data Availability

The data presented in this study are available on request from the corresponding author.

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
