# Peer review of "Capillary Western Immunoassay Optimization of Estrogen Related Factors in Human Subcutaneous Adipose Tissue"

_mps, 2022, doi:10.3390/mps5020034_

Round 1

Reviewer 1 Report

The topic of this manuscript falls within the scope of Methods and Protocols Journal. The topic of the manuscript is very interesting, relevant, and original.

The Authors have presented sufficient data. The appropriate tables and figures have been provided. The article is easy to read and logically structured.  The methods are adequately described. The Authors used appropriate statistic methods. The conclusions are consistent with presented evidence and arguments.

There are some comments in the reviewer opinion which should be taken under consideration by the Authors:

  1. Please cite the newest papers in this field:

Dilated Blood and Lymphatic Microvessels, Angiogenesis, Increased Macrophages, and Adipocyte Hypertrophy in Lipedema Thigh Skin and Fat Tissue   https://doi.org/10.1155/2019/8747461

 The Role of Obesity-Induced Perivascular Adipose Tissue (PVAT) Dysfunction in Vascular Homeostasis. Nutrients. 2021;13(11):3843.

Key signaling networks are dysregulated in patients with the adipose tissue disorder, lipedema. Int J Obes (2021). https://doi.org/10.1038/s41366-021-01002-1

  1. Please add the limitations of the study

Reviewer 2 Report

GENERAL COMMENTS

This is a very technical manuscript on an interesting determination, that perfectly suits the journal’s scope. It is well structured and easy to follow. Since my expertise is not on the technology applied, I will focus on the more translational and clinical aspects.  In this regard, the manuscript may benefit from considering the following points:

Page 2, line 67: formulate a hypothesis at the end of the Introduction, before the aims, instead of in page 5 at the beginning of the Results’s section.

Page 9: It may be useful and more informative to indicate efficacy and specificity of each of the proteins studied.

Discussion: add a comment on potential effects of specific pathophysiological conditions like, for example, obesity – BMI is only mentioned in the last sentence of the Discussion, but the potential implications are not discussed.

In line with the previous point, a further aspect to be contemplated is the potential relevance of lipolytical differences among patients in relation to analyzing the plausible impact of the lipid rich nature of adipose tissue. It would be worthwhile mentioning this given the involvement of aquaporin 7 in lipolytic rate and body weight control (Frühbeck G. Obesity: aquaporin enters the picture. Nature. 2005 Nov 24;438(7067):436-7).

Author Response

Reviewer 2 -

Reviewer 2 Comment 1:  Page 2, line 67: formulate a hypothesis at the end of the Introduction, before the aims, instead of in page 5 at the beginning of the Results’s section. 

Author Response 1:  This is moved.

Reviewer 2 Comment 2:  Page 9: It may be useful and more informative to indicate efficacy and specificity of each of the proteins studied. 

Author Response 2:  We are unsure what the reviewer is referring to, thus did not expand on this subject.

Reviewer 2 Comment 3:  Discussion: add a comment on potential effects of specific pathophysiological conditions like, for example, obesity – BMI is only mentioned in the last sentence of the Discussion, but the potential implications are not discussed. 

Author Response 3:  We believe we addressed this with the addition to the intro.

Reviewer 2 Comment 4:  In line with the previous point, a further aspect to be contemplated is the potential relevance of lipolytical differences among patients in relation to analyzing the plausible impact of the lipid rich nature of adipose tissue. It would be worthwhile mentioning this given the involvement of aquaporin 7 in lipolytic rate and body weight control (Frühbeck G. Obesity: aquaporin enters the picture. Nature. 2005 Nov 24;438(7067):436-7). 

Author Response 4:  We were unsure how to include this, thus we did not.